Can high-intensity interval training and small-sided games be effective for improving physical fitness after detraining? A parallel study design in youth male soccer players

http://orcid.org/0000-0001-9813-2842 Clemente Filipe Manuel 1 2 3 filipe.clemente5@gmail.com
http://orcid.org/0000-0003-0609-0601 Soylu Yusuf 4
http://orcid.org/0000-0002-2933-6937 Arslan Ersan 5
Kilit Bulent 6
Garrett Joel 7
van den Hoek Daniel 7
http://orcid.org/0000-0003-4100-8765 Badicu Georgian 8
http://orcid.org/0000-0002-1772-1272 Filipa Silva Ana 1 3 9
1 Escola Superior Desporto e Lazer, Instituto Politécnico de Viana do Castelo, Rua Escola Industrial e Comercial de Nun’Álvares, Viana do Castelo, Portugal , Melgaço , Portugal
2 Instituto de Telecomunicações, Delegação da Covilhã, Lisboa, Portugal , Lisboa , Portugal
3 Research Center in Sports Performance, Recreation, Innovation and Technology (SPRINT), Melgaço, Portugal , Melgaço , Portugal
4 Faculty of Sport Sciences, Tokat Gaziosmanpasa University, Tokat, Turkey , Yokat , Turkey
5 Faculty of Sport Sciences, Tokat Gaziosmanpasa University, Tokat, Turkey , Tokat , Turkey
6 Faculty of Sport Sciences, Tekirdag Namik Kemal University, Tekirdag, Turkey , Tekirdag , Turkey
7 Australian Catholic University, School of Behavioural and Health Sciences, Brisbane, Queensland, Australia , Queensland , Australia
8 Department of Physical Education and Special Motricity, Faculty of Physical Education and Mountain Sports, Transilvania University of Braşov, Braşov, Romania , Brasov , Romania
9 The Research Centre in Sports Sciences, Health Sciences and Human Development (CIDESD), Vila Real, Portugal , Vila Real , Portugal
García-Ramos Amador
Electronic publication date: 2022 Jul 1
Publication date: 2022
Volume: 10
Electronic Location ID: e13514
Received 2022 Feb 28; Accepted 2022 May 9
Copyright: © 2022 Clemente et al.
Copyright year: 2022
Copyright holder: Clemente et al.
License: This is an open access article distributed under the terms of the Creative Commons Attribution License, which permits unrestricted use, distribution, reproduction and adaptation in any medium and for any purpose provided that it is properly attributed. For attribution, the original author(s), title, publication source (PeerJ) and either DOI or URL of the article must be cited.
License URL: https://creativecommons.org/licenses/by/4.0/

Keywords: Football, Physical fitness, Athletic performance, High-intensity interval training, Human physical condition

Funding: Fundação para a Ciência e Tecnologia/Ministério da Ciência Tecnologia e Ensino Superior EU funds UIDB/50008/2020 This work is funded by Fundação para a Ciência e Tecnologia/ Ministério da Ciência, Tecnologia e Ensino Superior through national funds and when applicable co-funded EU funds under the project UIDB/50008/2020. The funders had no role in study design, data collection and analysis, decision to publish, or preparation of the manuscript.

==============================
Background

The aim of this study was two-fold: (i) analyze the within-group physical fitness adaptations promoted by a detraining period (4 weeks) followed by an intervention period (4 weeks) using small-sided games (SSGs) or running-based high intensity interval training (HIIT); and (ii) analyze the between-group differences aiming to identify the effectiveness of each training intervention on the physical fitness of youth male soccer players.

Methods

This study followed a randomized parallel study design. Forty male soccer players (age: 16.4 ± 0.5 years old) were assessed three times: (i) baseline; (ii) after 4 weeks of detraining; and (iii) after a retraining period of 4 weeks. After returning from detraining, players were randomized to an SSG-based training intervention (n = 20) or running-based HIIT (n = 20). Interventions lasted 4 weeks, with a training frequency of three sessions per week. At all timepoints, players were assessed by: (i) anthropometry (height, body mass, fat mass (FM)), countermovement jump (CMJ), standing broad jump (SBJ), triple hop jump (THJ), linear sprint test (5-, 10-, and 20-m), zig-zag test with (ZZwB) and without (ZZwoB) ball, three corner run test (3CRT), Y-balance test and the Yo-Yo intermittent recovery test level 1 (YYIRT). Mixed ANOVA (time * group) was conducted for testing interactions between the three timepoints of repeated measures and the two groups. Effect size (ES) for pairwise comparisons was calculated using Cohen’s.

Results

Between-group analysis revealed significantly smaller SBJ (t = −2.424, p = 0.020, d = −0.767 small ES) and THT (t = −4.353, p < 0.001, d = −1.376 large ES) in the SSG group after the retraining period. At the same time, SSG presented significantly greater FM after retraining compared to HIIT (t = 3.736, p < 0.001, d = 1.181 large ES). Additionally, SSG had significantly smaller values than HIIT in the ZZwB (t = −3.645, p < 0.001, d = −1.153 large ES), but greater times in the ZZwoB (t = 2.679, p = 0.011, d = 0.847 large ES) and 3CRT (t = 3.126, p = 0.003, d = 0.989 large ES).

Conclusions

Although SSG and HIIT interventions improved physical fitness outcomes after a period of detraining, they were not able to effectively restore body composition, CMJ, 20-m sprint, ZZwB, and YYIRT compared with the baseline assessments (before detraining). Only HIIT was significantly effective for restoring SBJ, short linear sprin speed, and change-of-direction compared with baseline. HIIT was also significantly better than SSG in improving SBJ and ZZwoB. Although the small sample, the non determination of maturation status and the need to be cautious regarding generalization, HIIT appears to be more beneficial than SSG after a detraining period for recovery of body composition and physical fitness qualities in this specific context of youth soccer players.

Introduction

Playing demands of soccer are mostly aerobic, interspersed with brief bouts of anaerobic activity (Stolen et al., 2005; Dolci et al., 2020). Thus, over the season players are generally able to improve or maintain their general aerobic and anaerobic fitness (Hammami et al., 2013). However, periods of detraining can occur, for example, in times of absence of practice (e.g., vacations) or during times of injury and illness (Requena et al., 2017; Suarez-Arrones et al., 2019) which can contribute to declines in physical fitness. Periods of detraining can be classified as either short or extended if they are shorter or longer than 4 weeks (Mujika & Padilla, 2000a). Detraining can result from a total training cessation or a substantial reduction in training intensity, volume or frequency. In regular competitive soccer schedules, training cessation caused by the off-season can last between 4 and 6 weeks (Jeong et al., 2011; Koundourakis et al., 2014). The reduction or cessation in training for this length of time can have substantial detrimental effects on body composition and physical fitness. For example, research exploring the effects of detraining in soccer players (Silva et al., 2016; Clemente, 2021) revealed significant increases in fat mass and decreases in aerobic fitness, strength, power, speed, and change-of-direction. Despite individualized off-season training programmes during these periods of training interruptions, it seems that these programmes only can ameliorate significant decrements in outcomes such as maximal oxygen uptake or repeated-sprint ability, while the remaining presents significant decrements (Clemente, 2021).

This has meant reversing the effects of detraining after periods of training cessation (i.e. the offseason period) has has become a priority of coaches and practitioners working within high performance soccer. Due to its ability in improving aerobic fitness, repeated sprint ability, and sprint performance, high-intensity interval training (HIIT) is commonly used in these periods of retraining (Clemente et al., 2021b). Although HIIT has shown to produce improvements in a variety of physical qualities, small-sided games (SSGs), that produces an intense physiological impact similar to HIIT (Boraczyński et al., 2022), is also commonly used when retratining athletes due to its similarity to match-play work-rates and its ability to also work on sport specific skills. This is based on the fact that SSGs highly stress the aerobic system specifically, aerobic power (Lacome et al., 2017).

Regardless of the method used, coaches must understand the differences in response to SSGs and HIIT to be able to periodize training programs and prepare athletes appropriately (Clemente, 2020). One issues with SSGs is that it does provide some intra- and inter-individual variability in high-intensity locomotor demands and possibly some under-stimulation of high-intensity running and sprinting (Clemente, 2020). However, SSGs provide an adequate stimulus for improving aerobic fitness (Moran et al., 2019) while facilitating technical/tactical stimuli (Clemente & Sarmento, 2020; Clemente et al., 2020, 2021c). A recent meta-analysis comparing SSGs with conventional endurance training, reported similar effects for positive aerobic fitness development (Moran et al., 2019). However, a meta-analysis comparing SGGs with running-based HIIT reported that HIIT was significantly better in improving sprinting, but not for improveing change-of-direction (COD) or vertical jump height (Clemente et al., 2021a). These systematic reviews present inconsistent evidence especially for youth players, since interventions very among adult professionals.

While recent research compares SSGs and HIIT for athletic development, there is limited evidence for their use in overcoming the effects of detraining. In fact, detraining is a topic that is present after every off-season or even after Christmas breaks or after returning to play. These cases represent examples of detraining that should be overcome using retraining strategies. For these moments, coaches should select the most appropriate methods to enhance players’ physical levels while ensuring an adequate technical/tactical stimulus for structuring the team’s model of play. This creates a need to decide which physiological based training should be used to enhance the physical fitness of soccer players without compromising the technical and tactical skills. Thus, the debate about the applicability of HIIT or SSG during the retraining period is still present and the majority of the studies comparing both do not focusing on this period of the training plan.

To the authors knowledge, no study has been conducted on SSG and its ability to enhance physical fitness after a detraining period or compared HIIT with SSGs for the purpose of retraining after a detraining period. This can provide coaches and practitioners information about the effectiveness of both in this specific context, helping in the decision making process to which methods may be more effective in youth soccer players. Thus, we aimed to compare the effects of SSGs and HIIT in youthsoccer players who had undergone 4 weeks of detraining. The aims of this study were two-fold: (i) analyze the within-group physical fitness adaptations promoted by a detraining period (4 weeks) followed by an intervention period (4 weeks) using SSGs or running-based HIIT; and (ii) analyze the between-group differences to identify the effectiveness of each training intervention on the physical fitness of youth male soccer players.

Materials and Methods

Study design

This study followed a randomized parallel study design. The protocol was approved by Faculty of Sport Sciences, Tokat Gaziosmanpasa University ethical committee with the code number E.4816-26439.

Setting

The study started on the 04/06/2018 and ended on the 23/08/2018. The study began a week after the last match of the season. The timeline of the study can be found in Fig. 1. Players were assessed three times (baseline, after a detraining period of 4 weeks, after a retraining period of 4 weeks). Per each period of assessment, 7 days were used to run the battery of tests. The first day of assessments occurred with a resting period of 72 h after the last training session/match. The assessments occurred between 8 and 10 a.m for all the assessments. The average temperature and relative humidity of the baseline assessments were 22 °C and 40%. For the period of assessment after detraining, the average temperature and relative humidity was 25 °C and 35%. Finally, for the assessments that occurred after retraining, the average temperature was 29 °C with a 32% relative humidity. No training sessions occurred between baseline and the detraining assessment period. During the detraining period, the players were asked to completely rest and avoid any kind of physical activity. In the retraining period, the players returned to their normal five training sessions per week, plus a match. The training sessions lasted on average 75–80 min. The specific intervention using SSG or HIIT occurred three times per week (Monday, Wednesday, Friday), starting immediately after a standardized warm-up protocol. In addition to their designed training for a total of 4 consecutive weeks, the players performed dynamic and static core strength training involving upper and lower body exercises two times per week (Tuesday and Thursday) (Arslan et al., 2021b).

Figure 1 Timeline of the study.

Participants

The sample size was estimated using the G * Power software (version 3.1). After adding a partial effect size of 0.2, a power of 0.8 and a p-value of 0.5 (two groups and three measurements) for a correlation of 0.5 a recommended total sample size of 10 was required. Inlcuded in this study were 40 players from two under-17 male soccer teams (characteristics of the included players can be found in Table 1), participating in the regional Turkish league. After returning from the detraining period, the players were randomly allocated to two groups, SSG and HIIT. Simple randomization was performed (Altman & Bland, 1999) to allocate each group 20 players. The eligibility criteria for inclusion of the players were: (i) players did not have an injury or illness longer than 1 day over the period of detraining and retraining; (ii) participants had an adherence greater than 90% of training sessions associated with the interventions; (iii) players participated in all the assessments and no missing data were observed. All the players were preliminary asked for their participation. The study design and protocol were verbally announced to players and their guardians. After their approval, the guardians voluntarily signed a free consent. The protocol followed the ethical standards for the study in humans, as suggested by Declaration of Helsinki.

Table 1 Participant’s characteristics (mean and standard-deviation) at the baseline.

	SSG Group	HIIT Group	Total	
Participants (n)	20	20	40	
Defenders (n)	7	8	15	
Midfielders (n)	9	8	17	
Attackers (n)	4	4	8	
Age (years)	16.3 ± 0.5	16.6 ± 0.5	16.4 ± 0.5	
Experience (years)	4.5 ± 0.6	4.7 ± 0.4	4.6 ± 0.5	
Height (cm)	174.8 ± 5.6	174.7 ± 6.2	174.7 ± 5.8	
Body mass (kg)	63.9 ± 6.4	62.6 ± 6.2	63.3 ± 6.2	
Body mass index (kg/m2)	21.0 ± 2.4	20.7 ± 3.0	20.8 ± 2.7	
Adherence (%)	100	100	100	

Assessment procedures

The physical test battery lasted 7 days for each period of assessment. The physical test battery was created based on the recommendations of previous research which suggested measure of aerobic capacity, acceleration, speed, speed-endurance, change-of-direction, strength, power and reactive strength (Turner et al., 2011). On day one of the assessments, the assessment order was the following: body composition, jumping and sprinting tests. On day three: agility tests and the 30–15 intermittent fitness test (VIFT). On day five: balance and three corner run test. On day seven: the Yo-Yo intermittent recovery test. Between tests, 5 min of rest were provided. With exception of the first day (which started with anthropometry and body composition assessments), the remaining assessments days started with a standardized warm-up protocol consisting of low-intensity running and stretching with basic passing. Players were informed to avoid meals 2 h before the assessments, and to avoid any stimulant drinks, drugs or supplementation.

Anthropometry and body composition

The stature was measured by using a stadiometer with an accuracy of 0.1 cm (SECA, Hamburg, Germany). Body mass and body fat percentage were measured using the bioelectrical impedance measurement (BC-418; Tanita, Tokyo, Japan). These measurements were performed with the players wearing no shoes and only light clothing in the morning before breakfast. The main outcomes extracted were the stature (height in centimeters), body mass (BM: kilograms), body mass index (BMI: kilograms/meters2) and fat mass (FM: percentage).

Countermovement jump (CMJ)

The players performed the countermovement jump (CMJ) with hands on the hips throughout the jump to minimize the contribution of the upper limbs. Jump performances were assessed using a portable force plate (Optojump; Microgate, Bolzano, Italy). The participants performed three trials, interspaced by 120 s of rest. The height of jump (CMJ: measured in centimeters) was taken, and the higher of jumps was used as main outcome. The ICC (intra-class correlation test) was 0.94 for the CMJ.

Standing broad jump (SBJ)

The participant stands behind the starting line and is instructed to push off vigorously and jump as far as possible. The participant were instructed to push off vigorously and jump as far as possible before landing feet together and upright. The distance was measured using a standard tape measure, which was perpendicular from the front of the start line to the posterior surface of the back heel at the landing (Castro-Piñero et al., 2009). The participants performed two trials, interspaced by 120 s of rest. The length of jump (SBJ: measured in centimeters) was taken, and the higher of jumps was used as main outcome. The ICC was 0.86 for the SBJ.

Triple jump hop test (THT)

Participants were required to reach the maximum distance with three consecutive hops without losing balance and touching the ground with any of their hands or the other leg (Hamilton et al., 2008). A standard tape measure was used for measuring the maximum distance in the THT. The participants performed two trials, interspaced by 120 s of rest. The length of jump (SBJ: measured in centimeters) was taken, and the higher of jumps was used as main outcome. The ICC was 0.83 for the THT.

Linear sprint test (ST)

Each player performed a linear 20-m sprint test (5-m, 10-m, and 20-m splits). The sprint test started with participants in a split position with the preferable leg forwards. The starting point position was 70 cm behind the first pair of photocells that marked the starting line. Four pairs of photocells were used (starting line, 5-m, 10-m and 20-m). The portable wireless photocell system (Witty; Microgate, Bolzano, Italy) was positioned at the player’s hip height. The participants performed two trials, interspaced by 120 s of rest. The time of the sprint distance (ST: measured in seconds) was taken, and the better of the two sprints was used as main outcome. The ICC was 0.87 for the ST.

Zig-Zag test with (ZZwB) and without (ZZwoB) ball

The agility performances of the players were evaluated using a ZZwB and ZZwoB test. The test consisting of four 5-m sections set out at 100° angles. This test was based on rapid deceleration, acceleration, and balance control required for a short running time (Little & Williams, 2005). The participants performed two trials, interspaced by 120 s of rest. The time of ZZwB and ZZwoB (measured in seconds) was taken, and the better of trials was used as main outcome. The ICC was 0.92 and 0.92 for the ZZwB and ZZwoB respectively.

Three cone shuttle drill test (3CRT)

The 3CRT was implemented as previous protocol (Rosch et al., 2000). The performance time was assessed using portable wireless photocell system (Witty; Microgate, Bolzano, Italy). The participants performed two trials, interspaced by 5 min of rest. The time of 3CRT (measured in seconds) was taken, and the better of trials was used as main outcome. The ICC was 0.94 for the 3CRT.

The 30–15 intermittent fitness test (VIFT)

To determine the running speed for the HIIT, the 30–15 intermittent fitness test, which has been shown to be reliable for HIIT prescription, was performed as previously described (Buchheit, 2008). The speed was noted as the final velocity obtained in the 30–15 intermittent fitness test (VIFT) during the last completed stage of the test. The VIFT was used to standardized the HIIT training.

The Yo-Yo Intermittent Recovery Test–level 1 (YYIRT)

To evaluate of aerobic capacity, The YYIRT was performed on a natural grass pitch according to procedures described previously (Bangsbo, Iaia & Krustrup, 2008). The final completed distance covered (measured in meters) at YYIRT was taken as the main outcome.

Y-balance test for right (YBT-R) and left (YBT-L) legs

The standardized leg length Y balance test was implemented for the anterior, posteromedial, and posterolateral directions. The protocol was implemented as in previous research (Plisky et al., 2006). The ICC was 0.96 for the YBT-R and YBT-L respectively. After the test, a composite score (CS) was calculated using the following formula (Filipa et al., 2010): CS = [(maximum anterior reach distance + maximum posteromedial reach distance + maximum posterolateral reach distance)/(leg length × 3)] × 100. The composite score (average) at YBT-R and YBT-L at YYIRT was taken as the main outcome.

Training intervention

The training interventions occurred on Monday, Wednesday and Friday of each week. The sessions started after a standardized warm-up protocol consisting of low-intensity running and stretching with basic passing. The sessions started at 17.30 pm, and during the period the average temperature was 29 °C and relative humidity of 35%. All players were familiar with all tests used in this study and were verbally encouraged by their team coach to exert maximal efforts during the testing and training sessions. All tests were performed on a synthetic grass pitch. The details of training intervention for each session can be observed in Table 2.

Table 2 Details of the training intervention.

Week	Sessions	SSG-group	Format/Pitch dimension (m)	Relative pitch size (m2)	PACES	RPE	ITL	HIIT-group	PACES	RPE	ITL	
Week 1	Session 1	2 × (2 × 2.30 min G-K), 2 min rest	2 vs. 2 + GK/20 × 18	75	27.6 ± 2.2	16.6 ± 0.9	198.6 ± 11.3	2 × (6 min of 15″-15″ at 90% of VIFT)	17.0 ± 1.7	17.5 ± 1.0	210.0 ± 12.0	
Session 2	
Session 3	
Week 2	Session 4	2 × (2 × 3 min S-G), 2 min rest	2 vs. 2 ball possession/20 × 15 m	29.1 ± 2.9	16.2 ± 1.1	226.8 ± 14.8	2 × (7 min of 15″-15″ at 90% of VIFT )	18.4 ± 1.6	17.7 ± 0.7	247.1 ± 10.4	
Session 5	
Session 6	
Week 3	Session 7	2 × (2 × 3.30 min POS), 2 min rest	30.4 ± 2.5	16.2 ± 1.1	258.4 ± 17.4	2 × (8 min of 15″-15″ at 90% of VIFT )	17.4 ± 1.6	18.3 ± 1.1	292.0 ± 17.1	
Session 8	
Session 9	
Week 4	Session 10	2 × (2 × 4 min F-G), 2 min rest	30.6 ± 2.0	16.5 ± 1.0	296.1 ± 18.0	2 × (9 min of 15″-15″ at 95% of VIFT )	18.6 ± 2.4	19.2 ± 0.6	345.6 ± 11.1	
Session 11	
Session 12	
Note:

SSG-group, small-sided games training; HIIT-group, high-intensity interval training; RPE, rating of perceived exertion; PACES, physical activity enjoyment scale; ITL, internal training load; VIFT, Maximum speed reached in the last stage of the 30–15 Intermittent Fitness Test; G-K, goalkeeper; S-G, small goal; POS, possession; F-G, free game.

Training drills and training intensity monitoring

Table 2 presents the characteristics of training intervention for both SSG and HIIT groups. The rating of perceived exertion (RPE) was obtained using the category ratio scale (6–20) to calculate the internal training load (ITL) immediately after the completion of each session (Foster et al., 2021). The scale was introduced at the beginning in order to familiarise the players. All players also completed a short form of the physical activity enjoyment scale (PACES). This scale includes five items scored on a 1–7 Likert scale and has been validated (Mirzeoğlu & Çoknaz, 2014) as a marker of enjoyment level for physical activity by Turkish youth (Arslan, Orer & Clemente, 2020).

Statistical procedures

Descriptive statistics are presented as mean and standard deviation. Exploratory analysis was conducted to check for possible significant outliers. Intra-class correlations were calculated for each outcome and reported in the methods to provide information about the reliability of the data. After no observed outliers, normality was tested using Kolmogorov-Smirnov test and homogeneity was tested using the Levene’s test. After confirmation of normality (p > 0.05) and homogeneity (p > 0.05) of the sample, mixed ANOVA (time * group) was conducted for testing interactions between the three moments of repeated measures and the two groups. Partial eta squared ( ηp2) was used to determine the effect size in the mixed ANOVA. Within-group analysis was conducted using repeated measures ANOVA and between-group analysis for each time point using the independent t-test. Bonferroni’s post-hoc test was used to determine the significance level in pairwise comparisons. Cohen’s d effect size (d) was used to determine the effect size in pairwise comparisons. Magnitude of effect size (d) was considered trivial (0.00–0.20), small (0.21–0.50), medium (0.51–0.80) and large (>0.81).All statistical procedures were executed in the SPSS (version 28.0.0.0, IBM, USA) for a p < 0.05.

Results

Mixed ANOVA tested the interactions between time (three periods of assessment) and groups (SSG and HIIT). Significant interactions (time * groups) were found for BM (F = 5.122; p = 0.016; ηp2 = 0.119), FM (F = 8.537; p = 0.002; ηp2 = 0.183), SBJ (F = 8.315; p = 0.002; ηp2 = 0.180), THT (F = 24.390; p < 0.001; ηp2 = 0.391), 10-mST (F = 11.561; p < 0.001; ηp2 = 0.233), 20-mST (F = 4.212; p = 0.028; ηp2 = 0.100), ZZwB (F = 8.962; p = 0.002; ηp2 = 0.191), ZZwoB (F = 15.341; p < 0.001; ηp2 = 0.288), 3CRT (F = 11.062; p < 0.001; ηp2 = 0.225), YYIRT (F = 8.152; p = 0.003; ηp2 = 0.177), VIFT (F = 8.154; p = 0.001; ηp2 = 0.306), YBT-R (F = 5.391; p = 0.008; ηp2 = 0.124) and YBT-L (F = 34.611; p < 0.001; ηp2 = 0.477). No significant interactions were found for CMJ (F = 2.846; p = 0.087; ηp2 = 0.070), and 5-mST (F = 2.214; p = 0.121; ηp2 = 0.055).

Table 3 presents the descriptive statistics of anthropometric and body composition outcomes. Within-group changes over the assessments revealed significant variations in both groups considering the outcomes of BM (SSG: F = 40.059, p < 0.001, ηp2 = 0.678; HIIT: F = 114.563, p < 0.001, ηp2 = 0.858) and FM (SSG: F = 40.966, p < 0.001, ηp2 = 0.683; HIIT: F = 84.393, p < 0.001, ηp2 = 0.816). Between-groups analysis revealed that SSG presented significantly greater FM after retraining compared to HIIT (t = 3.736, p < 0.001, d = 1.181).

Table 3 Descriptive statistics (mean and standard deviation) of anthropometric and body composition outcomes in the three assessment moments.

	SSG-group	SSG-group	SSG-group	SSG-group	HIIT-group	HIIT-group	HIIT-group	HIIT-group				
Outcome	Baseline	After detraining	After retraining	Within-group	Baseline	After detraining	After retraining	Within-group	Between-group (baseline)	Between-group (after detraining)	Between-group (after retraining)	
BM (kg)	63.9 ± 6.4b,c	66.4 ± 6.5a,c	65.2 ± 6.4a,b	F = 40.059 p < 0.001* ηp2 = 0.678	62.6 ± 6.2b,c	65.3 ± 6.6a,c	63.1 ± 6.3a,b	F = 114.563 p < 0.001* ηp2 = 0.858	t = 0.629 p = 0.533 d = 0.199	t = 0.528 p = 0.601 d = 0.167	t = 1.056 p = 0.298 d = 0.334	
%change	NA	+3.9%	−1.8%	NA	NA	+4.3%	−3.4%	NA	NA	NA	NA	
FM (%)	12.7 ± 1.5b,c	14.7 ± 1.3a,c	13.8 ± 0.9a,b	F = 40.966 p < 0.001* ηp2 = 0.683	12.2 ± 1.1b,c	14.9 ± 1.4a,c	12.7 ± 0.9a,b	F = 84.393 p < 0.001* ηp2 = 0.816	t = 1.219 p = 0.230 d = 0.386	t=−0.444 p=0.659 d=−0.140	t = 3.736 p < 0.001* d = 1.181	
%change	NA	+15.7%	−6.1%	NA	NA	+22.1%	−14.8%	NA	NA	NA	NA	
Notes:

a Significant different (p < 0.05) from baseline.

b Significant different (p < 0.005) from after detraining.

c Significant different (p < 0.005) from after retraining.

* Significant different.

BM, body mass; FM, fat mass; NA, not applicable; %change, represents the percentage of change regarding the immediate previous assessment.

Table 4 presents the descriptive statistics of jumping outcomes. Within-group changes over the assessments revealed significant variations in both groups considering the outcomes of CMJ (SSG: F = 97.122, p < 0.001, ηp2 = 0.836; HIIT: F = 112.919, p < 0.001, ηp2 = 0.856), SBJ (SSG: F = 71.395, p < 0.001, ηp2 = 0.790; HIIT: F = 109.321, p < 0.001, ηp2 = 0.852) and THT (SSG: F = 100.892, p < 0.001, ηp2 = 0.842; HIIT: F = 128.233, p < 0.001, ηp2 = 0.871). Between-group analysis revealed significant smaller SBJ (t = −2.424, p = 0.020, d = −0.767) and THT (t = −4.353, p < 0.001, d = −1.376) in the SSG group in the after retraining.

Table 4 Descriptive statistics (mean and standard deviation) of jumping outcomes in the three assessment moments.

	SSG-group	SSG-group	SSG-group	SSG-group	HIIT-group	HIIT-group	HIIT-group	HIIT-group				
Outcome	Baseline	After detraining	After retraining	Within-group	Baseline	After detraining	After retraining	Within-group	Between-group (baseline)	Between-group (after detraining)	Between-group (after retraining)	
CMJ (cm)	43.4 ± 4.5b,c	38.6 ± 3.6a,c	41.5 ± 4.1a,b	F = 97.122 p < 0.001* ηp2 = 0.836	43.2 ± 3.7b,c	38.8 ± 3.5a,c	42.4 ± 3.6a,b	F = 112.919 p < 0.001* ηp2 = 0.856	t = 0.134 p = 0.894 d = 0.043	t = −0.112 p = 0.911 d = −0.035	t = −0.738 p = 0.465 d = −0.234	
%change	NA	−11.1%	+7.5%	NA	NA	−10.2%	9.3%	NA	NA	NA	NA	
SBJ (cm)	219.1 ± 15.8b,c	190.5 ± 9.6a,c	213.8 ± 13.3a,b	F = 71.395 p < 0.001* ηp2 = 0.790	218.5 ± 15.8b	185.1 ± 14.1a,c	223.5 ± 12.1b	F = 109.321 p < 0.001* ηp2 = 0.852	t = 0.130 p = 0.897 d = 0.041	t = 1.415 p = 0.165 d = 0.447	t = −2.424 p = 0.020* d = −0.767	
%change	NA	−13.1%	+12.2%	NA	NA	−15.3%	+20.7%	NA	NA	NA	NA	
THT (cm)	611.7 ± 19.9b,c	564.7 ± 19.4a,c	582.8 ± 16.7a,b	F = 100.892 p < 0.001* ηp2 = 0.842	618.3 ± 26.2b	563.3 ± 38.0a,c	615.5 ± 29.2b	F = 128.233 p < 0.001* ηp2 = 0.871	t = −0.897 p = 0.375 d = −0.284	t = 0.152 p = 0.880 d = 0.048	t=−4.353 p < 0.001* d = −1.376	
%change	NA	−7.7%	+3.2%	NA	NA	−8.9%	+9.3%	NA	NA	NA	NA	
Notes:

a Significant different (p < 0.005) from baseline.

b Significant different (p < 0.005) from after detraining.

c Significant different (p < 0.05) from after retraining.

* Significant different;

CMJ, countermovement jump; SBJ, standing broad jump; THT, triple hop test; NA, not applicable; %change, represents the percentage of change regarding the immediate previous assessment.

Table 5 presents the descriptive statistics of sprinting, change-of-direction and agility outcomes. Within-group changes over the assessments revealed significant variations in both groups considering the outcomes of 5-mST (SSG: F = 22.758, p < 0.001, ηp2 = 0.545; HIIT: F = 20.621, p < 0.001, ηp2 = 0.520), 10-mST (SSG: F = 44.218, p < 0.001, ηp2 = 0.699; HIIT: F = 42.860, p < 0.001, ηp2 = 0.693), 20-mST (SSG: F = 38.612, p < 0.001, ηp2 = 0.670; HIIT: F = 85.011, p < 0.001, ηp2 = 0.817), ZZwB (SSG: F = 89.017, p < 0.001, ηp2 = 0.824; HIIT: F = 33.849, p < 0.001, ηp2 = 0.640), ZZwoB (SSG: F = 82.33, p < 0.001, ηp2 = 0.813; HIIT: F = 169.831, p < 0.001, ηp2 = 0.899) and 3CRT (SSG: F = 55.174, p < 0.001, ηp2 = 0.744; HIIT: F = 58.612, p < 0.001, ηp2 = 0.755). Between-groups analysis revealed that in the assessment after retraining the SSG-group had significant smaller values than HIIT in the ZZwB (t = −3.645, p < 0.001, d = −1.153), but was greater in the ZzwoB (t = 2.679, p = 0.011, d = 0.847) and 3CRT (t = 3.126, p = 0.003, d = 0.989).

Table 5 Descriptive statistics (mean and standard deviation) of sprint, change-of-direction and agility outcomes in the three assessment moments.

	SSG-group	SSG-group	SSG-group	SSG-group	HIIT-group	HIIT-group	HIIT-group	HIIT-group				
Outcome	Baseline	After detraining	After retraining	Within-group	Baseline	After detraining	After retraining	Within-group	Between-group (baseline)	Between-group (after detraining)	Between-group (after retraining)	
5-mST (s)	0.89 ± 0.05b,c	1.01 ± 0.05a,c	0.95 ± 0.05a,b	F = 22.758 p < 0.001* ηp2 = 0.545	0.94 ± 0.04b	1.04 ± 0.05a,c	0.97 ± 0.04b	F = 20.621 p < 0.001* ηp2 = 0.520	t = −3.865 p < 0.001* d = −1.222	t = −1.994 p = 0.053 d = −0.631	t = −1.125 p = 0.267 d = −0.356	
%change	NA	+13.5%	−5.9%	NA	NA	+10.6%	−6.7%	NA	NA	NA	NA	
10-mST (s)	1.55 ± 0.09b,c	1.74 ± 0.05a,c	1.66 ± 0.07a,b	F = 44.218 p < 0.001* ηp2 = 0.699	1.66 ± 0.06b	1.79 ± 0.06a,c	1.67 ± 0.05b	F = 42.860 p < 0.001* ηp2 = 0.693	t = −4.329 p < 0.001 d = −1.369	t = −1.955 p = 0.058 d = −0.618	t = 0.473 p = 0.639 d = 0.150	
%change	NA	+12.3%	−4.6%	NA	NA	+7.8%	−6.7%	NA	NA	NA	NA	
20-mST (s)	2.74 ± 0.17b,c	3.01 ± 0.14a,c	2.93 ± 0.14a,b	F = 38.612 p < 0.001* ηp2 = 0.670	2.87 ± 0.07b,c	3.10 ± 0.11a,c	2.95 ± 0.06a,b	F = 85.011 p < 0.001* ηp2 = 0.817	t = −3.096 p = 0.005* d = −0.979	t = −1.789 p = 0.082 d = −0.566	t = −0.583 p = 0.565 d = −0.184	
%change	NA	+9.9%	−2.7%	NA	NA	+8.0%	−4.8%	NA	NA	NA	NA	
ZZwB (s)	6.28 ± 0.28b,c	6.92 ± 0.36a,c	6.38 ± 0.28a,b	F = 89.017 p < 0.001* ηp2 = 0.824	6.40 ± 0.38b,c	6.96 ± 0.43a,c	6.77 ± 0.39a,b	F = 33.849 p < 0.001* ηp2 = 0.640	t = −0.989 p = 0.329 d = −0.313	t = −0.320 p = 0.750 d = −0.101	t = −3.645 p < 0.001* d = −1.153	
%change	NA	+10.2%	−7.8%	NA	NA	+8.7%	−2.7%	NA	NA	NA	NA	
ZzwoB (s)	5.13 ± 0.15b,c	5.59 ± 0.28a,c	5.40 ± 0.22a,b	F = 82.33 p < 0.001* ηp2 = 0.813	5.18 ± 0.21b,c	5.62 ± 0.21a,c	5.24 ± 0.19a,b	F = 169.831 p < 0.001* ηp2 = 0.899	t = −0.589 p = 0.559 d = −0.186	t = −0.522 p = 0.605 d = −0.165	t = 2.679 p = 0.011* d = 0.847	
%change	NA	+9.0%	−3.4%	NA	NA	+8.5%	−6.8%	NA	NA	NA	NA	
3CRT (s)	33.2 ± 1.2b,c	35.1 ± 1.3a,c	34.3 ± 1.3a,b	F = 55.174 p < 0.001* ηp2= 0.744	33.1 ± 1.2b	35.1 ± 1.3a,c	33.1 ± 1.2b	F = 58.612 p < 0.001* ηp2 = 0.755	t = 0.415 p = 0.681 d = 0.131	t = −0.012 p = 0.990 d = −0.004	t = 3.126 p = 0.003* d = 0.989	
%change	NA	+5.7%	−2.3%	NA	NA	+6.0%	−5.7%	NA	NA	NA	NA	
Notes:

a Significant different (p < 0.005) from baseline.

b Significant different (p < 0.005) from after detraining.

c Significant different (p < 0.05) from after retraining.

* Statistical significant at p < 0.05.

ST, linear sprint test; ZZwB, Zig-Zag test with ball; ZzwoB, Zig-Zag test without ball; 3CRT, three corner run test; NA, not applicable; %change, represents the percentage of change regarding the immediate previous assessment.

Table 6 presents the descriptive statistics of aerobic and balance testing outcomes. Within-group changes over the assessments revealed significant variations in both groups considering the outcomes of YYIRT (SSG: F = 175.897, p < 0.001, ηp2 = 0.903; HIIT: F = 248.193, p < 0.001, ηp2 = 0.929), VIFT (SSG: F = 445.165, p < 0.001, ηp2 = 0.959; HIIT: F = 358.149, p < 0.001, ηp2 = 0.950), YBT-R (SSG: F = 264.557, p < 0.001, ηp2 = 0.933; HIIT: F = 160.211, p < 0.001, ηp2 = 0.894) and YBT-L (SSG: F = 342.710, p < 0.001, ηp2 = 0.947; HIIT: F = 155.889, p < 0.001, ηp2 = 0.891).

Table 6 Descriptive statistics (mean and standard deviation) of aerobic and balance outcomes in the three assessment moments.

	SSG-group	SSG-group	SSG-group	SSG-group	HIIT-group	HIIT-group	HIIT-group	HIIT-group				
Outcome	Baseline	After detraining	After retraining	Within-group	Baseline	After detraining	After retraining	Within-group	Between-group (baseline)	Between-group (after detraining)	Between-group (after retraining)	
YYIRT (m)	1,738.0 ± 243.1b,c	1,331.0 ± 202.2a,c	1,568.0 ± 213.4a,b	F = 175.897 p < 0.001* ηp2 = 0.903	1,786.0 ± 259.4b,c	1,344.0 ± 198.6a,c	1,700.0 ± 247.3a,b	F = 248.193 p < 0.001* ηp2 = 0.929	t = −0.604 p = 0.550 d = −0.191	t = 0.025 p = 0.839 d = −0.065	t = −1.807 p = 0.079 d = −0.572	
%change	NA	−23.4%	+17.8%	NA	NA	−24.7%	+26.5%	NA	NA	NA	NA	
VIFT (km/h)	19.4 ± 0.9b,c	16.0 ± 0.5a,c	18.2 ± 0.6a,b	F = 445.165 p < 0.001* ηp2=0.959	19.6 ± 1.5b,c	16.0 ± 1.2a,c	18.8 ± 1.5a,b	F = 358.149 p < 0.001* ηp2 = 0.950	t = −0.689 p = 0.495 d = −0.218	t < 0.001 p => 0.999 d < 0.001	t = −1.571 p = 0.125 d = −0.497	
%change	NA	−17.5%	+13.8%	NA	NA	−18.4%	+17.5%	NA	NA	NA	NA	
YBT-R (cm)	113.5 ± 5.9b,c	99.6 ± 5.2a,c	108.5 ± 4.6a,b	F = 264.557 p < 0.001* ηp2 = 0.933	110.3 ± 7.3b,c	98.6 ± 6.8a,c	108.3 ± 8.0a,b	F = 160.211 p < 0.001* ηp2 = 0.894	t = 1.503 p = 0.141 d = 0.475	t = 0.548 p = 0.587 d = 0.173	t = 0.096 p = 0.924 d = 0.030	
%change	NA	−12.2%	+8.9%	NA	NA	−10.6%	+9.8%	NA	NA	NA	NA	
YBT-L (cm)	115.3 ± 5.1b,c	97.0 ± 5.6a,c	107.1 ± 5.1a,b	F = 342.710 p < 0.001* ηp2 = 0.947	109.0 ± 8.1b	97.9 ± 8.5a,c	107.9 ± 8.0b	F = 155.889 p < 0.001* ηp2 = 0.891	t = 2.897 p = 0.006* d = 0.916	t = −0.408 p = 0.686 d = −0.129	t = −0.365 p = 0.717 d = −0.115	
%change	NA	−15.9%	+10.4%	NA	NA	−10.2%	+10.2%	NA	NA	NA	NA	
Notes:

a significant different (p < 0.05) from baseline.

b significant different (p < 0.005) from after detraining.

c significant different (p < 0.005) from after retraining.

* Statistical significant at p < 0.05.

YYIRT, Yo-Yo Intermittent Recovery Test–level 1; VIFT, final velocity at 30–15 Intermittent Fitness test; YBT-R, y-balance test with right foo; YBT-L, y-balance test with left foot; NA, not applicable; %change, represents the percentage of change regarding the immediate previous assessment.

The SSG group had a greater SBJ (Fig. 2A) at baseline compared to after detraining (+28.6 cm; p < 0.001) and after retraining (+5.4 cm; p = 0.012). At baseline, the HIIT group had a significantly longer SBJ than after detraining (+33.35 cm; p < 0.001), but not retraining (−5.050 cm; p = 0.176. Considering the THT (Fig. 2B), the SSG-group had a greater jump at baseline compared to after detraining (+47.0 cm; p < 0.001) and after retraining (+28.9 cm; p < 0.001), while in the HIIT-group, THT was greater at baseline than after detraining (+55.1 cm; p < 0.001) but not retraining.

Figure 2 Variations of (A) standing broad jump, (B) triple hop jump test, (C) Zig-Zag with ball, (D) Zig-Zag without ball, (E) three cone test run.

The SSG group reported a better time while executing the ZZwB (Fig. 2C) at baseline in comparison to after detraining (−0.64 s; p < 0.001) and retraining (−0.10 s; p = 0.002). In the HIIT group at baseline, players had a significantly faster time at baseline than after detraining (−0.575 s; p < 0.001) and retraining (−0.385 s; p < 0.001). Regarding the ZzwoB (Fig. 2D), the SSG-group had a faster time at baseline in comparison to after detraining (−0.45 s; p < 0.001) and retraining (−0.28 s; p = 0.002). In the HIIT group, players were significantly faster at baseline than after detraining (−0.46 s; p < 0.001) and retraining (−0.07 s; p = 0.025). Finally, the SSG group had a faster time while executing the 3CRT (Fig. 2E) at baseline in comparison to after detraining (−1.88 s; p < 0.001) and retraining (−1.04 s; p = 0.002). The HIIT group were faster at baseline than after detraining (−2.040 s; p < 0.001) but not retraining (0.025 s; p > 0.999).

Figure 3 presents the average intensities (ITL) reported by players in each week and overall. The SSG training was significantly less intense than HIIT in week 1 (−0.95 A.U.; t = −3.089; p = 0.004), week 2 (−1.45 A.U.; t = −5.016; p < 0.001), week 3 (−2.10 A.U.; t = −6.150; p < 0.001) and week 4 (−2.75 A.U.; t = −10.483; p < 0.001) and average of all weeks (−1.81 A.U.; t = −11.687; p < 0.001).

Figure 3 Descriptive statistics (mean and standard-deviation) of rate of perceived exertion scores for each week (average).

Discussion

The 4-week detraining period had a significant detrimental impact on body composition and physical fitness (e.g., aerobic, jumping, linear sprint, change-of-direction, agility and balance) for youth male soccer players. The 4-week retraining period using either SSG or HIIT was not able to restore vertical jump, 20-m sprint, agility with ball, and aerobic fitness to baseline levels but they were able to improve these qualities over a 4-week intervention period. Interestinly, while neither method was able to return most qualities to baseline, HIIT was effective for restoring linear sprint at 5 and 10-m, horizontal jump, and change-of-direction (3CRT) relative to baseline measures. Between-group comparisons revealed that in the post-retraining period, HIIT was significantly better at improving FM, SBJ, THT, ZzwoB and 3CRT, while SSG was only significantly better in improving ZZwB.

Training cessation or a significant decrease in training intensity, volume, or frequency results in detrimental effects on body composition (e.g., increase in fat mass, decrease on lean mass) and physical fitness (e.g., decreases in aerobic fitness, strength, and power, and agility) (Mujika & Padilla, 2000b). Our results revealed a significant impairment over a 4-week detraining period in fat mass, aerobic fitness, vertical and horizontal jumping, linear sprinting, change-of-direction and balance. These results are in line with those reported in previous systematic reviews conducted in youth and adults soccer players (Silva et al., 2016; Clemente, 2021). These impairment can be caused, among others, by the decreases in muscle capillary density and oxidative enzymes resulting in reduced mitochondrial adenosine triphosphate production after training cessation, which in combination with a reduced arterial-venous oxygen difference may explain the decrements in aerobic fitness observed in our study (Mujika & Padilla, 2001). Moreover, although short term detraining does not significantly change muscle fiber distribution, it may influence fiber cross-sectional area. This may compromise force production, accompanied by a reduction in electromyographical activity, which can be observed in reduced performance of power-related actions like vertical and horizontal jumping, sprinting or change-of-direction (Mujika & Padilla, 2001).

Improvements promoted by SSG and HIIT over 5 to 6 weeks, independently or combined, on aerobic fitness have been confirmed previously in youth male soccer players (Arslan, Orer & Clemente, 2020; Arslan et al., 2021b, 2021a). SSGs and HIIT predominantly sustain efforts above 85% of maximum heart rate, which signals the development of aerobic power, is one of the causes for explaining the improvement of aerobic fitness after these drills (Castagna et al., 2011). Given that SSG and HIIT require a considerable number of direction changes, accelerations and decelerations, it is logical that we observed favourable changes when introduced after a detraining period (Young & Rogers, 2014; Mota et al., 2021). Additionally, because of the strain and mechanical stimulus in SSG and HIIT, we would expect to observe improvements in neuromuscular performance (as seen over the intervention period), primarily based on the stretching-shortening cycle (Buchheit & Laursen, 2013). However, only HIIT restored linear sprint at five and 10-m, horizontal jump, and change-of-direction (3CRT) to baseline levels. This is likely linked to the high neuromuscular strain occurring in high-intensity running with change-of-direction which implies a mechanical strain caused by the deceleration and acceleration. Accelerations and decelerations performed in a structured delivery may help promote a stimulus that allows a faster adaptation on players than in SSG (Buchheit & Laursen, 2013). In our study, the 4 weeks of SSGs were not enough to restore any of physical fitness outcomes considering the baseline levels (before detraining). This evidence can be justified by the limited formats used which implies a smaller locomotor stimulus which possibly can be not enough for a proper adaptation. Possibly, a longer intervention should be needed for completing restoration to baseline levels using SSGs (Clemente et al., 2022).

This study also showed that following the retraining intervention, the HIIT group had significantly better horizontal jump and change-of-direction without a ball performance than SSG group. On the other hand, SSG resulted in significantly better agility with a ball. These results align with a previous research (Chaouachi et al., 2014) conducted on youth players, which compared multidirectional sprints and SSG effects on change-of-direction and jumping performance after 6-week period intervention. Due to the specificity of training, it is logical that SSGs should better restore agility with a ball, while HIIT is more effective for improving agility without a ball (Chaouachi et al., 2014). Similarly, HIIT results in players achieving greater running speeds which may explain the greater restorative effect of this modality for linear running and horizontal jumping (Sales et al., 2018).

Overall, our results show that SSG and HIIT can improve physical fitness after a detraining period. We observed that both the SSGs and HIIT groups significantly improved fat mass, aerobic fitness, vertical and horizontal jump, linear sprint, change-of-direction, agility with ball and balance over the retraining intervention period. Retraining after detraining aims to restore a player’s physical fitness without exposing them to a greater risk of injury. Two main aims of retraining must be considered: (i) attenuating the adverse effects of detraining; and (ii) restoring performance to baseline levels. The findings of this study suggest that coaches may use either training method to mitigate the adverse effects of detraining.

Study limitations and future research

One of the current research study limitations is the absence of a control group not exposed to a training intervention. Considering this as a potential bias, we cannot firmly state that improvements in physical fitness after retraining resulted only from the use of interventions. The remaining training sessions also likley played a role (e.g. the strength sessions). Future research should add a control group with no intervention to identify how much SSG or HIIT can contribute independent of the remaining training session. This study also only utilized a 4-week retraining period which may not be sufficient time for the development of a number of physical qualities. While it may highlight that HIIT interventions could be more benifical in the initial training periods after the off-season, longer intervention periods are needed to understand the longer term effects of utilizing each method. Finally, in the context of the small sample size, this can affect the generalization of evidence. Additionally, the absence of data regarding the maturation status should be also faced as a possible bias for generalization. Thus, future research should be conducted with more participants and consider different competitive levels, while monitoring maturation status.

Generalisability, and practical implications

The context of data collection and the small sample does not provide enough ability to generalize the evidence. Thus, any conclusion should be circumscribed to the current case. In the current population, it was observed that SSG and HIIT were enough for improving body composition and physical fitness of youth male soccer players after a 4 week detraining period. However, HIIT results in faster and bigger improvement in horizontal jump and change-of-direction without the ball relative to SSG. Moreover, HIIT appears to be more effective than SSG in restoring player fitness outcomes to baseline levels during a 4-week retraining intervention. Coaches must consider the balance between restoring fitness outcomes with football’s tactical and skill-based demands. As such, a combination of SSG and HIIT should be considered. Conceivably, HIIT can be an used as an approach to ensure a standardized stimulus, while SSGs can be used to generate a drill closer to the model of play of the team. Additionally, coaches may also consider using a combination of both to try to take advantage of the strengths of the methods.

Conclusions

This study revealed that a 4-week detraining period significantly negatively affects body composition and physical fitness. Although limitations of the small sample and the absence of a control of maturational status, the data revealed that HIIT is more effective than SSG for the restoration of body composition and physical fitness to baseline levels after a detraining period. Although 4 weeks of retraining with SSGs and HIIT significantly improved body composition and physical fitness after detraining, it was not enough to return to baseline levels for body composition, vertical jump, 20-m sprint, agility with ball, and aerobic fitness. Moreover, only HIIT was effective for restoring linear sprint at 5 and 10-m, horizontal jump, and change-of-direction (3CRT) to baseline levels. Comparisons between groups also allowed us to conclude that HIIT appeared to be superior in improving horizontal jump, change-of-direction without the ball, and fat mass, while SSGs was superior for improving agility with the ball.

Supplemental Information

Supplemental Information 1 Raw data.

Click here for additional data file.

Additional Information and Declarations

Competing Interests

Author Contributions

Human Ethics

Data Availability

Filipe M. Clemente and Georgian Badicu are Academic Editors for PeerJ.

Filipe Manuel Clemente conceived and designed the experiments, analyzed the data, prepared figures and/or tables, authored or reviewed drafts of the article, and approved the final draft.

Yusuf Soylu conceived and designed the experiments, performed the experiments, authored or reviewed drafts of the article, and approved the final draft.

Ersan Arslan conceived and designed the experiments, performed the experiments, authored or reviewed drafts of the article, and approved the final draft.

Bulent Kilit conceived and designed the experiments, performed the experiments, authored or reviewed drafts of the article, and approved the final draft.

Joel Garrett conceived and designed the experiments, authored or reviewed drafts of the article, and approved the final draft.

Daniel van den Hoek conceived and designed the experiments, authored or reviewed drafts of the article, and approved the final draft.

Georgian Badicu conceived and designed the experiments, authored or reviewed drafts of the article, and approved the final draft.

Ana Filipa Silva conceived and designed the experiments, authored or reviewed drafts of the article, and approved the final draft.

The following information was supplied relating to ethical approvals (i.e., approving body and any reference numbers):

The Faculty of Sport Sciences, Tokat Gaziosmanpasa University approved the study (4816-26439).

The following information was supplied regarding data availability:

The raw data is available in the Supplementary File.

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
