# Peer review of "Can high-intensity interval training and small-sided games be effective for improving physical fitness after detraining? A parallel study design in youth male soccer players"

_PeerJ, doi:10.7717/peerj.13514_

## Round 0.1 · original submission · Major Revisions

Reviewers found the topic of the study interesting. However, there are some aspects related to the analysis made that should be modified or clarified.

Reviewer 1 ·

Basic reporting

Introduction clearly highlights the area but the literature sources used are limited to a handful of authors. The hypothesis for the study is clear and article structure is fine.

Experimental design

It is an interesitng question that the research design seeks to answer. As a true randomisation was carried out there are differences between the groups at baseline which limits the usefulness when looking at different training modalities. A clear description of all the tests which would allow for replication of the work.

Validity of the findings

Given that there are differences between the groups at baseline then there is a problem with appliying the findings. How can you assume one training modality is better than another when they are not starting from the same point. The authors need to address this problem as the discussion is meaningless at the moment. I would suggest either normalising the values to baseline for each group (% change or fold change are common for this) or to change the statistical analysis to an ANCOVA and using baseline test as a covariate. Until the analysis is doen correctly then the discussion is pointless.

Reviewer 2 ·

Basic reporting

- The authors need to work with a fluent English speaker/writer to correct grammatical and punctuation errors throughout the manuscript.
- The title is not well formulated.
- Why you choose this comparison (SSG vs. HIIT) in the detraining period.
- Is there a difference in results between youth players and professional players?
- The problematic is not clear. Thanks to clear it.
- More ideas are needed in this part.
- L78- L81: Thanks to reformulate this paragraph

Experimental design

- Why you choose this test battery not another ?
- Which method used to control training intensity ?
- Assessment procedures is not very clear.

Validity of the findings

- I think that figure 1 and 2 are not very clear to the reader.
- Table 5 is very complex. So, i suggest trying to simplify information in this table.
- L430-L433: This part need to be reformulated
- L454-L457: Thanks to revise this part
- At the bottom of the discussion the authors should:
*Discuss limitations of the study, taking into account sources of potential bias or imprecision. Discuss both direction and magnitude of any potential bias
*Discuss the generalisability (external validity) of the study results
*Discuss the practical implications and future research

Reviewer 3 ·

Basic reporting

Nice effort by the research team. The article addresses a great topic and a clear need for research within this area. The information you provided within the literature is relevant and laid out nicely leading into your purpose statement. However, there are a few edits which need to be addressed to ensure the article is sound and ready for publication. My main concern are the originality, the methodology, and the conclusions made. You may need to reformulate the title, to explain more the analysis data, and to soften the conclusions based on the number of subjects and the age of the subjects. The research team makes some pretty broad conclusions that probably need to be softened.

Experimental design

acceptable but needs a lot of explanation and clearly

Validity of the findings

nice conclusions

Additional comments

General Comments:
Nice effort by the research team. The article addresses a great topic and a clear need for research within this area. The information you provided within the literature is relevant and laid out nicely leading into your purpose statement. However, there are a few edits which need to be addressed to ensure the article is sound and ready for publication. My main concern are the originality, the methodology, and the conclusions made. You may need to reformulate the title, to explain more the analysis data, and to soften the conclusions based on the number of subjects and the age of the subjects. The research team makes some pretty broad conclusions that probably need to be softened.
Title:
Rephrase the title (change or modify the title)
Abstract:
L44: include maturity offset I the abstract.
L47: YoYo intermittent recovery test (which level 1 or 2).
Include the statistical tool used in this investigation
Include the intervention of d (small; medium or large)
Keywords:
Delete physical fitness
Change human physical condition
Introduction:
There are many interesting ideas in the introduction but they are not well organized and well classified.
What is the originality of this work ???? highlight it in the introduction
L84-85: i believe that this sentence should be placed at the beginning of the introduction
L90 : remove the parenthesis
L99: Clemente et al ?????
Check the references
Materials and Methods:
Study design:
The second assessment (after 4 weeks detraining), may be include injury in agility or sprint test????
In detraining period; what they have achieved the players during this period, how did you control that ?????? if not mentioned as a limit of this work
Participants:
Both teams training the same training (the load,….)????
More information about both teams (i.e., training weekly; friend game….)
Assessment procedures:
Include the CCI and CV% for each test (sprint, change of direction, jump, balance)
Zig-Zag test: did they control the technique of driving the ball ????? if yes, added description (inside, outside of the foot)
Anthropometry and body composition: include the maturity status of the players (The maturity status of each participant was calculated as a maturity offset (Mirwald RL, Baxter-Jones AD, Bailey DA, Beunen GP. An assessment of maturity from anthropometric measurements. Med Sci Sports Exerc 2002 Apr;34(4):689-94. PubMed PMID: 11932580. Epub 2002/04/05. eng.
):
Maturity Offset = −9.236 + 0.000278 leg length × sitting body height −0.001663 age × leg length + 0.007216 age × sitting body height + 0.02292 body weight × body height (years).
)
Training intervention:
this section needs a lot of explanation and clarification
SSG
• number of players (1vs1; 2vs2; 3vs3; 4vs4)
• number of the touch the ball (how many times to hit the ball: 1, 2, 3, gold games free)
• number of set of SSG is very small……
HIIT
• number of set
• with or without change of direction
Results:
Summarize the result section.
The figures are well presented.
Put the first and the third table in a single table.
Add the dimentions of the piutch in the table 2.
Correct p values (p<0.05 instead of p<0.005) in the tables.
L311 and 314: more explication ??? why you used d and n2p
Discussion:
The discussion requires a lot of explanation
L441: how did you control the heart rate (i.e., 85% of maximum heart rate)
L451-452: more explication (i.e., neuromuscular strain)
Be broad in the discussion i.e., when citing a reference add some detail of the reference such as the population, the period of the intervention…..
L487-495: conclusions or limitations
Conclusions:
Reformulate conclusions

I recommend this manuscript be revised and resubmitted due to inaccurate formatting, grammar, and depth and relevancy of literature review. In the current form, this paper does not convey an adequate message to the readership of the Peer J. Please check journal format for the Peer J and take a look at the writing tips so that each section of the paper is intact. The paper is not properly formatted for the journal. As written, this study does not meet the standards of a scholarly journal such as Peer J. The authors are encouraged to gain clarity on a) intention b) purpose c) format d) meaningful contributions to literature. I strongly encourage the authors to further pursue this area and resubmit as a new manuscript with major revisions.

Annotated reviews are not available for download in order to protect the identity of reviewers who chose to remain anonymous.

---

## Round 0.2 · accepted · Accept

Congratulations on meeting the high standard publication of PeerJ!

Reviewer 1 ·

Basic reporting

no comment

Experimental design

no comment

Validity of the findings

no comment

Additional comments

The authors have addressed concerns and the manuscript now reaches the desired level

Reviewer 2 ·

Basic reporting

No comment

Experimental design

No comment

Validity of the findings

No comment

Additional comments

No comment

Reviewer 3 ·

Basic reporting

Nice job addressing the requested revisions. The manuscript is much improved and ready for the next phase after one critical read and edit. Thank you for all of your hard work and attention to detail.

Experimental design

Nice job addressing the requested revisions. The manuscript is much improved and ready for the next phase after one critical read and edit. Thank you for all of your hard work and attention to detail.

Validity of the findings

Nice job addressing the requested revisions. The manuscript is much improved and ready for the next phase after one critical read and edit. Thank you for all of your hard work and attention to detail.

Additional comments

Nice job addressing the requested revisions. The manuscript is much improved and ready for the next phase after one critical read and edit. Thank you for all of your hard work and attention to detail.